# Identification of the Ricin-B-Lectin LdRBLk in the Colorado Potato Beetle and an Analysis of Its Expression in Response to Fungal Infections

**DOI:** 10.3390/jof7050364

**Published:** 2021-05-06

**Authors:** Ulyana N. Rotskaya, Vadim Yu. Kryukov, Elena Kosman, Maksim Tyurin, Viktor V. Glupov

**Affiliations:** Institute of Systematics and Ecology of Animals of Siberian Branch of the Russian Academy of Science, Frunze Str. 11, 630091 Novosibirsk, Russia; vereshchagina86@gmail.com (E.K.); maktolt@mail.ru (M.T.); skif@eco.nsc.ru (V.V.G.)

**Keywords:** antifungal peptide, attacin, C-type lectins, heat shock protein, domain pfam00652, RCA

## Abstract

Ricin-B-lectins (RBLs) have been identified in many groups of organisms, including coleopterans insects, particularly the Colorado potato beetle *Leptinotarsa decemlineata* (LdRBLs). We hypothesized that one of these LdRBLs (LdRBLk) may be involved in the immune response to fungal infections. We performed a theoretical analysis of the structure of this protein. Additionally, the expression levels of the *LdRBlk* gene were measured in *L. decemlineata* in response to infections with the fungi *Metarhizium robertsii* and *Beauveria bassiana*. The expression levels of *LdRBlk* in the *L. decemlineata* cuticle and fat body were increased in response to both infections. The induction of *LdRBlk* expression was dependent on the susceptibility of larvae to the fungi. Upregulation of the *LdRBlk* gene was also observed in response to other stresses, particularly thermal burns. Elevation of *LdRBlk* expression was frequently observed to be correlated with the expression of the antimicrobial peptide *attacin* but was not correlated with *hsp90* regulation. Commercially available β-lectin of ricin from *Ricinus*
*communis* was observed to inhibit the germination of conidia of the fungi. We suggest that LdRBLk is involved in antifungal immune responses in the Colorado potato beetle, either exerting fungicidal properties directly or acting as a modulator of the immune response.

## 1. Introduction

C-type lectins (CTLs) are a large class of Ca^2+^-dependent lectins with various functions; in particular, these proteins are involved in the immune responses of plants and animals [1]. For example, among vertebrates, the participation of CTLs in the initiation of the antifungal immune response cascade in HIV-positive *Homo sapiens* has been widely studied [2,3]. Among invertebrates, there are examples of increased *OrCTL* gene expression in the ganglion and hepatopancreas of the sea mollusk *Onchidium reevesii* following infection with the bacteria *Vibrio harveyi* and *V. parahaemolyticu* [4]. One of the CTLs of marine invertebrates, *Cucumaria echinata*, which uses galactose and *N*-acetylgalactosamine (Gal/GalNAc) as substrates in the presence of Ca^2+^ ions, exhibits hemolytic activity towards human and rabbit erythrocytes [5]. CTLs have been characterized in insects of different orders, particularly Lepidoptera, Diptera, Coleoptera and Hymenoptera [6,7,8,9]. In species of Lepidoptera (e.g., *Bombyx mori*, *Hyphantria cunea* and *Manduca sexta*), the ability of CTLs to participate in the immune response was studied by the following mechanisms: recognition of bacterial and yeast lipopolysaccharides [10,11,12], agglutination of bacteria and yeasts in the presence of Ca^2+^ ions [13,14] and induction of the phenoloxidase cascade [15,16,17]. The phenoloxidase cascade plays an important role in the immune response during the penetration of pathogens through the cuticle and encapsulation in hemolymph by the formation of melanin and production of highly toxic compounds, including semiquinone radicals [18].

The wide class of CTLs includes proteins with ricin-B-lectin domains (RBLs), which were first discovered as one of the ricin plant toxin domains in the castor bean plant (*Ricinus communis*) [19]. Due to the folding of the tertiary structure of this domain, its full name is ricin-type β-trefoil lectin domain (pfam00652) [20]. Currently, more than 1400 sequences of this lectin have been recorded for bacteria, fungi, plants and insects (orders Coleoptera and Blattodea) in GenBank databases. This domain is also present in the Colorado potato beetle *Leptinotarsa decemlineata* (BioProject *L. decemlineata* RefSeq genome sequencing, PRJNA420356, 2017), an economically significant species that is considered to be the primary pest affecting potato crops.

The agglutinating properties of RBLs were previously studied for the fungi *Hericium erinaceus*, *Stereum hirsutum* [21] and *Clitocybe nebularis* [22]. The authors showed the ability of RBLs of these fungi to recognize highly fucosylated *N*-glycans within glycoproteins on the surface of human colorectal carcinoma cells (for RBLs of both fungal species). In addition, RBL from *C. nebularis* binds to N,N′-diacetyllactosediamine in glycoproteins on the surface of human leukemic T cells.

In coleopterans, RBLs usually present several orthologs in one species; for example, in the Colorado potato beetle, it is found in several copies (NCBI accession numbers XP_023029739.1, XP_023025121.1, XP_023029750.1 and XP_023029736.1). We hypothesized that the peptide containing the RBL domain detected in the GEEF01084863.1 transcriptomic contig (LdRBLk) may play a role in the insect immune response similar to that of CTLs; in other words, RBLs may recognize and bind carbohydrates of the cell walls of fungi and bacteria and exert an inhibitory (possibly toxic) effect on microorganisms.

In the present work, to understand whether the studied peptide LdRBLk has the ability to participate in immune processes, we analyzed its protein structure. A phylogenetic analysis of the amino acid sequence of LdRBLk was also performed to clarify its relationship with other RBL peptides found in *L. decemlineata*, other insects and other organisms containing the RBL domain. Next, using qPCR, we analyzed the influence of fungal infections on the expression of LdRBLk in Colorado potato beetle larvae. We investigated two different ascomycetes, *Beauveria bassiana* (Cordycipitaceae) and *Metarhizium robertsii* (Clavicipitaceae), as well as larval stages with different susceptibility to fungal infection. We also compared the expression of *LdRBlk* under various cuticular injuries and fungal infection. To understand the relationship between the expression of *LdRBlk* and other immune- and stress-related genes, we studied the expression of the heat shock protein HSP90, which is a folding protein that can be considered as a stress marker [23], and the antimicrobial peptide attacin, which acts against Gram-negative bacteria [24]. To test the effect of RBL on fungal growth, we performed in vitro tests on the effect of commercially available *R. communis* RBL on the germination of *M. robertsii* and *B. bassiana* conidia on artificial media.

## 2. Material and Methods

### 2.1. Identification and Bioinformatic Analysis of the LdRBlk Peptide

The uncharacterized RNA sequence containing the RBL domain is located at the sequence of the Transcriptome Shotgun Assembly for *L. decemlineata* contig GEEF01084863.1 [25]. Reading frame (ORF), primary, secondary and tertiary structures were analyzed online using the ExPASy.org translation tool [26], SOPMA [27,28] and SWISS-MODEL [29]. To investigate whether this domain belongs to signal peptides, the online program SignalP-5.0 Server was utilized [30,31]. The presence of transmembrane regions was determined by the online program TMpred [32,33]. Additionally, the secondary structure, tertiary structure and ligand binding site were analyzed using the I-TASSER online program [34,35,36,37]. The hydrophilicity and hydrophobicity of the LdRBLk peptide were analyzed using the online ProtScale program using the Kyte–Doolittle and Hopp–Woods algorithms [38,39].

Based on the work of Nashman and Crowell [40], the mutation rate (μ) was estimated as µ = *κ*/(2 × *t* × *L*_bp_), where *κ* is the number of point substitutions per site, *t* is the time since the populations of *L. decemlineata* were derived and measured in generations and *L*_bp_ is the length of the investigated site in base pairs.

### 2.2. Phylogenetic Analysis

The phylogenetic alignment of amino acid sequences of five groups of lectins was performed to reveal the relationship of LdRBLk with other *L. decemlineata* peptides containing the RBL lectin domain (pfam00652) and the probable orthology of LdRBLk and the RBL of ricin toxin of *R. communis*. The first group was lectins of lactobacteria, which are gut associates of *L. decemlineata* [41], and bifidobacteria as a reference. The second group was fungal lectins from *Rhizoctonia solani* because it is a potato pathogen. The third group included lectins of insects from the order Coleoptera, especially *L. decemlineata* (LdRBLk, XP_023029736.1, XP_023029739.1 and XP_023029750.1), *Tribolium castaneum* and *Asbolus verrucosus*. The fourth group was plant lectins in which the ricin-type β-trefoil lectin domain (pfam00652) was combined with the domain of the glycoside hydrolase 5 family (GH5), including 3 representatives of the Solanaceae family (*S. tuberosum*, *S. chilense* and *Capsicum chinense*) and a glycoside hydrolase-associated RBL from *R. communis*. The fifth group of lectins with domain architecture analogous to the RBL of ricin toxin of *R. communis* (RIP domain and doubled ricin-B-lectin domain) was supplemented with orthologous lectins from *Viscum album*, *V. coloratum* and *Camellia sinensis* protein toxins. In total, 25 amino acid sequences with a final length of 632 amino acids were analyzed.

The searches of LdRBLk orthologs were performed using the online BLASTp program [42] and utilizing protein domains in the NCBI [43]. Multiple alignment of amino acid sequences was performed using the CLUSTAL algorithm in the MAFFT online program (v7.471) [44,45] (Watherhouse, 2009). The phylogenetic relationships among the species were reconstructed using the Maximum Likelihood method implemented in the MEGA X program [46]. The WAG with Freqs model (WAG + F) was recognized as the best model for describing the nature of amino acid sequence substitutions (minimal BIC was 26590.987 and minimal AICc was 26262.149) with 500 replicates bootstrap [47].

### 2.3. Fungi and Insects

We used the strains *B. bassiana* (Sar-31) and *M. robertsii* (P-72) from the collection of microorganisms of the Institute of Systematics and Ecology of Animals of the Siberian Branch of the Russian Academy of Science. Fungi were cultivated on two autoclaved millet [48] following collection of conidia by a soil sieve and storage at 4 °C. Conidia were suspended in a water–Tween 20 solution (0.05%), and their concentration was determined using a Neubauer hemocytometer.

Colorado potato beetle larvae were collected from a domestic field near Karasuk town (Russia, Novosibirsk region, 53°43′05.4″ N 77°38′14.4″ E). Larvae were maintained in the laboratory under a temperature of 25 °C and a photoperiod of 14:10 (light:dark), and they were fed *Solanum tuberosum* foliage.

### 2.4. Effect of B. bassiana and M. robertsii Infection

In this experiment, we evaluated the effect of fungal infections caused by *B. bassiana* and *M. robertsii* on the expression level of *LdRBlk*, heat shock protein (*hsp90*) and the antimicrobial protein *attacin* in *L. decemlineata* larvae. Larvae (2–4 h post-molt in instar IV) were infected by being dipped in a suspension of fungal conidia at a concentration of 5 × 10^6^ conidia/mL for 15 s. In the control, insects were treated with a water–Tween solution. The larvae were housed in 300-mL ventilated plastic containers and fed daily with potato leaves. Mortality assays were performed in 5 replicates (1 replicate = 10 larvae) and recorded for 8 days. At 48 h after infection, cuticles and fat bodies were collected. The cuticle was cleared of the fat body, and the fat body was separated from the cuticle, intestines and Malpighian vessels in cold phosphate buffer and frozen in liquid nitrogen. Five or six biological replicates (1 replicate = 5 fat bodies and 1 replicate = 5 cuticles) were used for analysis of gene expression.

### 2.5. Effect of Different Susceptibility to Fungi

Larval stages with different susceptibility to *M. robertsii* were experimentally analyzed. The highly susceptible stage (4 h post-molt in instar IV) and the resistant stage that finished feeding (86 h post-molt in instar IV) were used [48]. Larvae were collected in the field and the stages were detected based on phenotypes and weight: 50 ± 1.3 mg for susceptible larvae, and 146 ± 4.3 mg for resistant larvae [48]. Larvae were infected by being dipped in *M. robertsii* suspension (5 × 10^6^ conidia/mL) and maintained as described above. Five replicates (1 replicate = 10 larvae) were used for the mortality assay. At 24 h and 72 h after infection, cuticles were collected and frozen in liquid nitrogen for analysis of the expression of *LdRBlk* and *hsp90* genes. Five or six biological replicates (1 replicate = 5 cuticle) were used for the analysis.

### 2.6. Effect of Cuticular Injuries

Larvae 4–6 h post-molt in instar IV were analyzed. Chemical burns were achieved by dropping 15% HCl onto the dorsal area (5 µL, 5 min exposure). Thermal burns were achieved by brief touching of the dorsal area of larvae with a hot needle syringe. Amputation of the distitarsus segment of the right front leg was employed as mechanical damage of the integuments. In this case, the motor activity of the larva either did not decrease or was restored after 5 min. *M. robertsii* fungi were applied topically; specifically, 5 µL of a fungal suspension (4 × 10^8^ conidia/mL) was dropped onto the dorsal area of the larvae. In the case of the control variant, 5 µL water was dropped onto the dorsal area. Larvae were maintained as described above. Five replicates (1 replicate = 10 larvae) were used for the mortality assay. At 24 h post-treatment, fat bodies and cuticles were collected and frozen in liquid nitrogen for analysis of changes in the expression of the *LdRBlk*, *hsp90* and *attacin* genes. Five or six biological replicates (1 replicate = 5 larvae) were used for analysis.

### 2.7. Sample Preparation and qPCR

All tissues were disrupted by pestle in liquid nitrogen after lyophilization at 800 mTorr for 24 h. Cuticle homogenization was developed in Lira solution (BioLabMix, Novosibirsk, Russia) by a FastPrep-24 glass bead homogenizer (MP Biomedical, Irvine, CA, USA). Fat body homogenization was performed in Lira solution by pipetting with a 1-mL pipette tip. Further isolation of total RNA was performed with Lira solution according to the manufacturer’s instructions. Purification of RNA from DNA contamination and conversion to cDNA were performed using DNase (Promega, Madison, WI, USA) and RevertAid (Fermentas, Vilnius, Lithuania), respectively. qPCR was performed using the HS-qPCR SYBR Blue (2×) kit (BioLabMix, Novosibirsk, Russia) on a CFX96 Touch instrument (Bio-Rad Laboratories, Inc., Hercules, CA, USA). The quality of the qPCR product was checked by agarose gel electrophoresis and was confirmed by sequencing in the SB RAS Genomics Core Facility (ICBFM SB RAS, Novosibirsk, Russia).

The relative change in the expression of genes of interest was calculated by the ∆∆Cq method using Bio-Rad CFX Manager software (Bio-Rad Laboratories, Inc., Hercules, CA, USA). The 60S ribosomal proteins L4 and L18 (*Rp4* and *Rp18*) and ADP-ribosylation factors 4 and 1 (*Arf2* and *Arf19*) were used as reference genes. The primers were synthesized by Biosynthesis (Koltsovo, Russia). The primer properties were verified by IDT OligoAnalyzer 3.1 software [49]. The sequences of primers and qPCR products are presented in the tables (Appendix A).

### 2.8. Germination Assay

In the assay, we used commercially available A β-lectin (RBL) from *R. communis* (castor bean) agglutinin RCA120 (#L7886, Sigma-Aldrich, Munich, Germany). Both proteins (LdRBLk and RBL) have the same ligands specificity: galactose and *N*-acetyl-d-galactosamine (see Section 3.6). A buffered aqueous solution RBL was added to modified Sabouraud broth (10.0 g dextrose, 2.5 g peptone and 2.5 g yeast extract) to final RBL concentrations of 100, 50, 25 and 12.5 µg/mL. Sterile water was added to media as a control. The media were aliquoted into 1.5-mL Eppendorf tubes in a volume of 500 µL. A 20-µL water–Tween conidial suspension (5 × 10^7^ conidia/mL) of *M. robertsii* and *B. bassiana* was added to the tubes. Three replicates for each concentration of RBL were used. The tubes were incubated horizontally at 25 °C and 190 rpm. After 36 h of incubation, the number of germinated and non-germinated conidia was counted under a light microscope. At least 10 fields of view with 100 conidia were inspected for each replicate.

### 2.9. Statistics

Differences in the mortality dynamics were assessed by the Kaplan–Meier survival analysis (log-rank test). Gene expression data had an abnormal distribution (Shapiro–Wilk test: *p* < 0.05) and were analyzed by Dunn’s test. Differences in the conidia germination rate were analyzed by the two-way ANOVA followed by Tukey’s post-hoc test. Differences were considered significant at *p* < 0.05. Data on the plots are presented as arithmetic means and standard errors (SE).

## 3. Results

### 3.1. Identification and Bioinformatic Analysis of the LdRBlk Peptide

The full-length *LdRBLk* gene was identified in the GEEF01084863.1 contig sequence (Transcriptome Shotgun Assembly database of NCBI). The open reading frame (ORF) of the peptide is 381 base pairs, encoding 127 amino acids (Figure 1, Appendix A). The molecular weight of the peptide is 14.033 kDa, its isoelectric point (pI) is 6.02, its instability index is 46.26 and its grand average of hydropathicity index (GRAVY) is −0.155.

Nucleotide differences were observed between data for the populations obtained from the south of Western Siberia and the Central Sands region of Wisconsin, represented in the TSA NCBI databases. Two synonymous substitutions were identified (transversion 168 A → C and transition 195 C → T) in the sequence of the 134-bp PCR product obtained by qPCR. At the same time, no single point mutation was detected in sequences of any other genes of interest or reference genes. Under the hypothetical condition of the divergence of the two populations approximately 160 years ago, with *L. decemlineata* undergoing three generations per year, the rate of mutation accumulation in this locus (134 bp) was approximately 1.55 × 10^−5^ per site per generation.

According to a study of the secondary structure of this domain using the SOPMA program, its secondary structure was 11.02% alpha helix, 40.94% extended strand, 14.96% beta turn and 33.07% random coil (Figure 2A). When the tertiary structure model was constructed, the model in which the peptide fit into a globular structure, with the N- and C-ends positioned next to each other, exhibited the greatest reliability. A model was chosen for which the values of the indexes of probability and reliability of existence were optimal: GMQE = 0.49, QMEAN = −3.18 (Figure 2B). The SWISS-MODEL program also determined that based on the primary structure, the LdRBLk peptide had the greatest tertiary-structure similarity with the HA1 (HA33) subcomponent *Clostridium botulinum* type C progenitor toxin (14.53%) and with the hemolytic lectin CEL-III (22.22%).

An evaluation of the probable ligand binding site by the I-TASSER program (Appendix A) showed that with the highest probability (C-score = 0.51), the target ligand for LdRBLk was *N*-acetyl-d-galactosamine (NGA) with the involvement of amino acid residues at positions 58, 59, 60, 69, 71, 74, 76 and 78 with cluster size 115 and PBD Hit 3aj5B (Figure 2C).

An analysis of the probability that the LdRBLk peptide belongs to signal peptides in the SignalP-5.0 program showed that: Signal peptide (Sec/SPI) was 0.0048%, and Other (the probability that this peptide has no function of a signal peptide) was 0.9952% (Appendix A). Thus, the LdRBLk cannot be considered as a signal peptide. Prediction of the presence of a transmembrane domain in the LdRBLk peptide in the TMHMM program showed that this domain was not a transmembrane peptide (Appendix A). The analysis of hydrophilic properties by the online program ProtScale using the Kyte–Doolittle (Appendix A) and Hopp–Woods (Appendix A) algorithms showed that the peptide had hydrophobic N- and C-termini, while in the region of amino acid residue 78 (asparagine (N)), there was a strong hydrophilic center (GDD**N**QQF).

### 3.2. Phylogenetic Analysis

The purpose of the phylogenetic analysis was to determine the phylogenetic relationships of the studied peptide LdRBLk with other ricin-B-lectins of the Colorado potato beetle, and to establish its relationship with the RBLs of *R. communis*.

It was determined that all four analyzed amino acid sequences of RBLs of the Colorado potato beetle were composed of one clade with a support level of 98%. This clade was within the clade for RBLs of the Coleoptera order with 89% support. The groups of plant protein toxins and plant hydroxyl hydrolases organized separate clades with support of 98 and 100%, respectively. The RBLs groups of bacteria and the fungi *R. solani* formed separate clades with 73 and 99% support, respectively (Figure 3). Within the clades, RBLs were similar to one another in domain architecture and structure organization, according to the classification for plant lectins proposed by Peumans and Van Damme [50].

At the same time, the studied peptide LdRBLk was not a close ortholog with RBL of the ricin toxin of *R. communis* and was in a clade with other lectins of *L. decemlineata* and other species of the order Coleoptera. The clade of plant protein toxin lectins containing RBL of *R. communis* was significantly different from the clade of RBLs included in the plant glycoside hydrolase 5 family (GH5) (Figure 3).

Multiple alignment of the amino acid sequences of LdRBLs (LdRBLk, XP_023029736.1, XP_023029739.1 and XP_023029750.1) obtained using the CLUSTAL algorithm in the MAFFT program showed low homology between *L. decemlineata* peptides (Appendix A). The largest difference was at the N-terminus. At the same time, in the region of the hydrophilic center and at the C-terminus, there were identical amino acids or amino acids of the same type for all four sequences. Particularly highly homologous was the region corresponding to amino acid residues 78–79 in the LdRBLk sequence (NQQFYINSDGTI), coinciding with the hydrophilic center. This result indirectly indicates that the hydrophilic highly homologous center is the conserved region of the LdRBL peptides. The flanking ends of the peptides are highly variable, especially the N-terminus.

### 3.3. Effect of B. bassiana and M. robertsii on the Expression of the LdRBlk, Attacin and hsp90 Genes

The mortality of larvae was 76 and 100% on the 8th day after treatment with *M. robertsii* and *B. bassiana*, respectively (Figure 4A). Mortality of control larvae was 20%. *B. bassiana* showed significantly higher virulence than *M. robertsii* (log-rank test: *χ**^2^* = 11.2, df = 1, *p* < 0.001).

Both fungal infections caused a 3.5- to 3.8-fold increase in the level of *LdRBlk* expression in the cuticle at 48 h after treatment (Dunn’s test, *p* < 0.009 versus control, Figure 4B). A greater than 20-fold increase in *LdRBlk* expression was observed in the fat body (Dunn’s test, *p* < 0.01 versus control).

In the cuticle of larvae infected with *B. bassiana*, we observed a 75-fold increase in the expression of the antimicrobial peptide *attacin* gene (*p* < 0.009 compared with the control, Figure 4C). After infection with *M. robertsii*, the expression of *attacin* was not different from that of the control (*p* = 0.1). In the fat body, a 400-fold increase in the expression of *attacin* was observed after treatment with *B. bassiana* (*p* < 0.0005 relative to the control). After treatment with *M. robertsii*, the increase was 30-fold and not significant (*p* = 0.07).

A weak (1.7-fold) but significant increase in the expression of *hsp90* was observed in the cuticle of insects infected with *B. bassiana* (*p* < 0.03, Figure 4D) but not *M. robertsii* (*p* = 0.78). The expression of *hsp90* in the fat body of infected insects did not change significantly.

### 3.4. Expression of the LdRBlk and hsp90 Genes in Larvae with Different Susceptibility to Fungi

The level of expression of *LdRBlk* and *hsp90* in the cuticle of different age groups of larvae (4 h and 86 h post-molt in instar IV), differing in susceptibility to fungal infection, was assessed. Susceptible larvae (4 h post-molt) died at a frequency of 100% on the 5th day post-treatment, while the death of resistant larvae (86 h post-molt) did not differ significantly from the control (Figure 5A).

It was shown that the expression of the *LdRBlk* gene in the cuticle depends on the degree of susceptibility to *M. robertsii*. In susceptible larvae, 11-fold and 26-fold increases in expression were observed 24 and 72 h after infection, respectively (*p* < 0.005 compared to control, Figure 5B). For unsusceptible larvae, a very weak and insignificant increase was registered (*p* > 0.18). The basal gene expression levels in different age groups of larvae were the same.

The basal expression level of the *hsp90* gene was lower in the recently molted larvae than in the elderly larvae and was especially pronounced after 72 h of incubation (Figure 5C). Significant changes in gene expression in response to fungal infection were observed only at 72 h after infection. In particular, a 2.1-fold increase in expression was detected in the susceptible group (4 h post-molt) (*p* = 0.02 compared to control), and a 1.7-fold decrease in expression was detected in the resistant group (86 h post-molt) (*p* = 0.03 relative to the control). No significant correlation between the level of *hsp90* and *LdRBlk* was observed (*r* < 0.26, *p* > 0.73).

### 3.5. Gene Expression after Various Injuries to Integuments

To determine whether the increase in *LdRBlk* expression was a specific response to fungal infection, we performed experiments on the regulation of this gene after various injuries to the cuticle. All larvae survived for 8 days after amputation of the distitarsus (Figure 6A). After chemical and thermal burns, low mortality (13–17%) was observed only on the first and second days. Topical treatment with *M. robertsii* resulted in 60% mortality on day 8.

In the cuticle, a significant 10-fold increase in the expression of *LdRBlk* was observed after thermal burns (*p* < 0.0002 compared to control, Figure 6B), and a 4.6-fold increase was detected after topical fungal treatment (*p* < 0.01 compared to control). An insignificant increase in *LdRBlk* expression was observed after chemical burn and amputation of the distitarsus (*p* = 0.07 and *p* = 0.8 compared to the control, respectively). In the fat body, a significant increase in *LdRBlk* expression was also registered after thermal burns (*p* < 0.001 compared with the control). Expression increased to the level of marginal significance after amputation (2.5-fold, *p* = 0.06 compared with the control). Other treatments (chemical burns and topical fungal infection) did not lead to changes in *LdRBlk* expression in the fat body (*p* > 0.6 compared to the control).

Expression of the antimicrobial peptide *attacin* gene in the cuticle increased 48-fold after chemical burns and 56-fold after thermal burns (*p* < 0.003 and *p* < 0.001, respectively, compared to the control, Figure 6C). In the case of distitarsus amputation, the expression of *attacin* in the cuticle increased insignificantly (4-fold, *p* = 0.2 compared to the control). In the fat body, a significant increase in expression was detected only after thermal burns (14-fold, *p* < 0.002 compared to control). The increase was insignificant in the case of chemical burns and amputation (4–5-fold, *p* = 0.1 compared with the control). Topical treatment with *M. robertsii* did not lead to a change in *attacin* gene expression in either tissue relative to the control (*p* > 0.7).

*hsp90* gene expression did not significantly change in the fat body and cuticle after chemical burns, thermal burns and amputations (*p* > 0.2 compared to the control, Figure 6D). In the case of fungal infection, there was a 1.4-fold decrease in expression in the cuticle and a 1.3-fold decrease in expression in the fat body (*p* < 0.004 and *p* = 0.1, respectively, compared to the control).

There was a significant correlation between the expression of *LdRBlk* and *attacin* in the fat body (r = 0.91, *p* = 0.03), but a significant correlation was not observed in the cuticle (r = 0.67, *p* = 0.22). No correlation was found between the expression of *LdRBlk* and *hsp90*, either in the fat body or in the cuticle (r = 0.06, *p* = 0.93 and r = 0.29, *p* = 0.64, respectively).

### 3.6. Effect of β-Lectin of Ricin (RBL) from R. communis on the Germination of M. robertsii and B. bassiana Conidia

We assayed the effect of β-lectin of ricin from *R.*
*communis* (RBL) on the germination of fungi since the LdRBLk peptide has the same ligand specificity as RBL (*N*-acetyl-d-galactosamine) (Appendix A). The dose-dependent inhibition of conidial germination under the influence of RBL was observed for *M. robertsii* and *B. bassiana* (concentration effect—F_4.20_ = 6568.3, *p* < 0.001, Figure 7). Fungi responded differently to different doses of RBL (the effect of the fungus—F_1.20_ = 2084.1, *p* < 0.001; interaction between factor concentration and fungus—F_4.20_ = 1493.0, *p* < 0.001). Inhibition of germination was observed at RBL concentrations of 50 or 100 μg/mL. For *B. bassiana*, the number of germinated conidia at an RBL concentration of 25 μg/mL and below was >95%, but at a concentration of 50 μg/mL, germination strongly decreased to 6.2% (Tukey’s test, *p* < 0.001). At 100 μg/mL, no germinated conidia of *B. bassiana* were observed. *M. robertsii* was more resistant to RBL. In particular, germination of >99% of conidia was observed at concentrations of 50 μg/mL and below, and strongly reduced germination was observed only at a concentration of 100 μg/mL (5.1%, *p* < 0.001).

## 4. Discussion

In this study, the protein structure and target ligand of LdRBLk, the ricin-B-lectin of the Colorado potato beetle, were established for the first time, indicating that this protein may play important roles in the antifungal defense of this insect. It has been shown that in response to fungal infections, the *LdRBlk* gene is upregulated in the cuticle and fat body of *L. decemlineata* larvae. The level of increased expression was observed to directly correlate with insect susceptibility to fungal infection. The change in *LdRBlk* gene expression was also determined to correlate with the regulation of the antibacterial peptide *attacin* gene but was not observed to be associated with the regulation of the heat shock protein *hsp90* gene. However, the upregulation of the *LdRBlk* gene was observed not only in response to fungal infections but also against other conditions associated with damage to the cuticle, particularly thermal burns.

Our data on the theoretical analysis of the structure of LdRBLk showed that this peptide contains only one β-lectin domain (pfam00652). This characteristic distinguishes this peptide from plant glycoside hydrolases, where the ricin-type β-trefoil lectin domain (pfam00652) is linked into one enzyme with the glycoside hydrolase domain. Moreover, this property distinguishes this peptide from plant protein toxins, such as ricin, abrin and their orthologs, where the pfam00652 domain is duplicated in one peptide and linked to the RIP (ribosome inactivated protein) domain [51]. This finding means that all of these plant lectins are chimerolectins by structure [50]. The organization of LdRBLk also differs from the domain architecture in bacteria and fungi, in which the pfam00652 domain is usually repeated two or three times [52].

In the secondary structure of the peptide, we detected alternation of extended strand-forming β-sheets with β-turns and random coils. This division of the ricin-type β-trefoil lectin domain into highly conserved subdomains (QxW)_3_ localized in the extended strand was described by Hazes et al. [53]. In this case, the β-turns and random coil sections are highly variable and may contain extended inserts (Appendix A).

Additionally, in the secondary structure of the peptide, we detected the hydrophilicity of the center and the hydrophobicity of its N- and C-termini. Thus, LdRBLk is an amphiphilic peptide. An amphiphilic structure is characteristic of antifungal peptides [54]. This structure helps these peptides bind more effectively to the surface of fungal cell membranes [55]. The LdRBLk peptide has a structure with β-sheets that can interact with the surface of the fungal membrane according to the “carpet” model [54]. With this interaction, the antifungal peptide does not integrate into the membrane of the fungus but forms a “carpet pattern” on its surface. In this case, the hydrophobic side of the carpet faces the fungal membrane, and the hydrophilic side has the opposite orientation. As a result, the phospholipid bilayer of the membrane is distorted, which is followed by cell disruption and lysis [56].

The region of the peptide at the 78th amino acid residue, asparagine (GDD**N**QQF), is simultaneously the most highly conserved, carries a strong hydrophilic center and participates in the binding of the target ligand at positions 74, 76 and 78. The presence of highly variable N- and C-termini in LdRBLk may also be associated with the participation of the peptide in insect innate immunity. The mechanism by which highly variable regions of protein molecules participate in the immune response is typical for vertebrate antibodies [57]. The level of mutagenesis in the *LdRBlk* gene (1.55 × 10^−5^ per site per generation) is high against the background of previously established values for the rate of mutagenesis for various organisms [40,58].

Analysis of the tertiary structure of LdRBLk indicates its similarity with the HA1 (HA33) subcomponent of the bacterium *C. botulinum* type C progenitor toxin (14.53%) and with hemolytic lectin from the marine invertebrate *C. echinata* CEL-III (22.22%). In addition, analysis of the tertiary structure indicates that LdRBLk possesses the highest degree of affinity for *N*-acetyl-d-galactosamine. These data are consistent with finding obtained by studies of the *C. botulinum* toxin, in which the HA1 subcomponent is the ricin-type β-trefoil lectin, which binds to galactose and *N*-acetyl-d-galactosamine [59,60]. The hemolytic lectin CEL-III from *C. echinata* is also an ortholog of the β-lectins of the plant toxins abrin and ricin and exhibits similar properties, including high hemagglutinating activity in relation to mammalian erythrocytes [61,62,63].

Phylogenetic analysis of the studied LdRBLk sequence relative to other Colorado potato beetle LdRBLs showed that they are orthologs. All four peptides follow the one domain–one peptide scheme. Phylogenetic analysis of amino acid sequences of RBLs for insects of the order Coleoptera, fungi *R. solani*, bacteria, plant glycoside hydrolases (GHs) and plant protein toxins, as expected, showed that all groups formed separate clades. This result was confirmed by works on plant lectins containing the ricin-type β-trefoil lectin domain in the plant toxin ricin and its homologs from *Abrus precatorius*, *Sambucus nigra* and *V. album* (in the RIP/ricin B/ricin B configuration) and in plant glycoside hydrolases (GH25 and GH5) (in the ricin B/GH configuration) [64,65]. These authors showed that the studied domains were distributed among separate clades, although they had a common orthological ancestor. It is possible that such a distribution by clades for all five groups is supported not only by the primary structure of the protein but also correlated with the tertiary structure—by variants of domain alignment in peptides. Unfortunately, at present, such studies on domain composition have been presented only for plant lectins [50,51,66].

We investigated the possible antifungal properties of a commercially available RBL from *R. communis* ricin toxin. Particularly, we showed that this toxin inhibits the germination of *B. bassiana* and *M. robertsii* conidia (Figure 7). The inhibitory activity of lectins in plant toxins (such as ricin, abrin and their homologs) against plant viruses and insect pests is well known [64,67,68,69,70]. The capability of RBL to bind with the carbohydrate epitopes of *B. bassiana* conidia, blastospores and strong binding with hemolymph-derived hyphal bodies was previously shown by Wanchoo and coworkers [71]. We hypothesize that the LdRBLk and RBL of *R. communis* that we investigated may have similar antifungal activity. Despite the low homology of the amino acid sequences of these proteins, their mechanism of action may be similar due to the preservation of the general domain structure (pfam00652) and their ligand specificity (galactose and *N*-acetyl-d-galactosamine) (Appendix A). The influence of RBL on conidia germination of *B. bassiana* and *M. robertsii* was different. These results have raised an additional question about interaction of RBL peptides with different species of entomopathogenic fungi.

In this study, we showed a significant increase in the expression of the *LdRBlk* gene in the *L. decemlineata* cuticle and fat body in response to fungal infections with *B. bassiana* and *M. robertsii*. This finding is notable because this lectin is not classified with peptides with previously established antifungal activity, such as cecropins and thanatin [72]. However, we cannot state that the action of this protein is highly specific against fungal infections. It is known that the homolog of this lectin (CCL2) in *Coprinopsis cinerea* fungi exhibits toxicity towards the nematode *Caenorhabditis elegans* and the fly *D. melanogaster* [73]. A homologous nematotoxic lectin was also found in the parasol mushroom *Macrolepiota procera* [74]. These studies indicate that ricin-like β-lectins may function not only against fungi but also in the defense of some fungal species against invertebrates.

Expression of the *LdRBlk* gene in the cuticle tissue in response to *M. robertsii* infection correlated directly with the susceptibility of *L. decemlineata* larvae to this fungus. This result was in keeping with previously obtained findings on the level of activation of other antifungal reactions (such as the melanization response and activation of glutathione-S-transferases) in response to *M. robertsii* infection in these susceptible and resistant stages of the Colorado potato beetle [48]. In particular, susceptible recently molted larvae have a stronger activation of these systems compared to resistant finished feeding larvae. Thus, the upregulation of *LdRBlk* may be an indicator of the acuity of fungal infection.

We observed an increase in *LdRBlk* expression not only during fungal infections but also after thermal burn of the cuticle. In addition, slight trends towards the upregulation of *LdRBlk* in the cuticle or in the fat body have been observed after chemical burns and amputation of the destitarsus. It is well-known that various stresses, such as heating, cooling or damage to the integument, stimulate the immune response, particularly leading to an increase in the expression of antibacterial and antifungal peptide genes. For example, it has been shown in the wax moth *Galleria mellonella* that short-term or prolonged cooling, as well as heating, lead to multiple increases in the antifungal peptides gallerimycin, galiomycin and some antibacterial peptides [75,76,77]. Mechanical damage of the cuticle led to a notable increase in *attacin* genes in the parasitoid *Nasonia vitripennis* [78]. These alterations may be related to the adaptive response under stresses when upregulation of AMP could occur even in the absence of pathogen infection. In addition, infection with various microorganisms, including fungi, could occur through cuticle damage, which may also lead to an increase in *LdRBlk* expression. Both explanations correspond to the simultaneous increase in the expression of *LdRBlk* and *attacin* observed in the experiment with stresses.

Notably, an increase in *attacin* gene expression was observed simultaneously with an increase in *LdRBlk* gene expression only under *B. bassiana* infection. In the case of *M. robertsii*, the change in *attacin* gene expression was not significant (Figure 4). Attacin is known to be active against Gram-negative bacteria [79]. We assume that a significant increase in *attacin* expression during infection with *B. bassiana* may be associated with the penetration of bacteria through the integument. It has been observed that an influx of Gram-negative bacteria in the cuticle and hemocoel of Colorado potato beetle larvae occurs during infection caused by *B. bassiana* [80,81]. Apparently, the significant upregulation of the *attacin* gene after infection with *B. bassiana* (but not *Metarhizium*) may be observed because *Beauveria* fungi are weaker antagonists of Gram-negative bacteria than *Metarhizium* [82].

Changes in the expression of the heat shock protein *hsp90* gene did not correlate with those for *LdRBlk* in any of the performed experiments. Previously, we also observed weak changes in *hsp90* gene expression after treatment of the Colorado potato beetle larvae with fungi and insecticides [83]. Notably, in experiments with different age groups, *hsp90* showed an increase in expression towards the end of larval development and the transition to the prepupal stage. Most likely, HSP90 is weakly associated with the stress response in the Colorado potato beetle and is more necessary for the restructuring of the body during metamorphosis as a folding protein.

## 5. Conclusions

We have shown for the first time that the ricin-B-lectin *LdRBlk* is expressed in the Colorado potato beetle in response to fungal infections with *B. bassiana* and *M. robertsii*, as well as in response to some stresses, such as thermal burns. The expression level of this gene depends on insect susceptibility to fungi and is correlated with the expression of other AMPs, such as attacin. Bioinformatic analysis of the structure of this protein indicates that the mechanism of the antifungal action of LdRBLk is most likely similar to those of other antifungal peptides [54], but requires further study. Orthologous proteins (such as the RBL of *R. communis*) are able to suppress the development of the fungi *B. bassiana* and *M. robertsii* in vitro. Subsequent studies may focus on (1) cloning the *LdRBlk* gene into a bacterial system for production and studying its effect on entomopathogenic fungi in vitro, (2) assessing the expression of ricin-B-lectins in response to fungal invasions in other organisms, particularly plants, (3) silencing or knocking down of *RBL* genes in insects and determining susceptibility to fungal infections, (4) assessing pathomorphological and stress responses in entomopathogenic fungi under the influence of RBLs and (5) analysis of the *LdRBlk* expression in different stages of the *L. decemlineata* life cycle, including eggs, pupae and adults, in response to different infections and stresses.

## Figures and Tables

**Figure 1 jof-07-00364-f001:**
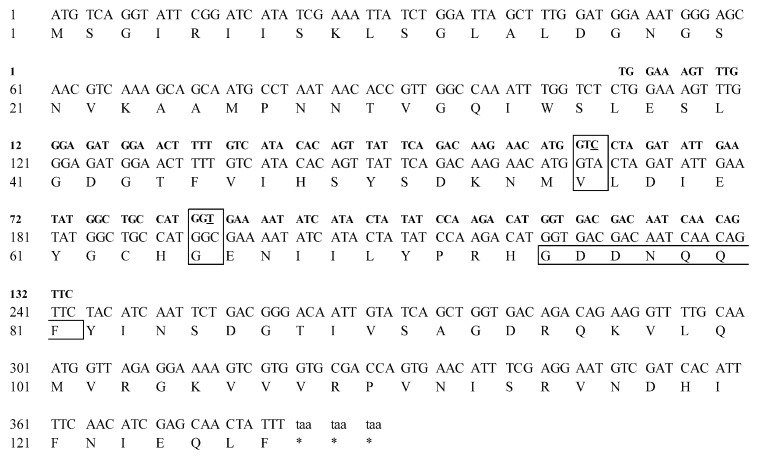
Amino acid sequence of the LdRBLk protein (**bottom line**) aligned with the nucleotide sequence of the *LdRBLk* gene from NCBI TSA data (**middle line**) and the nucleotide sequence of the PCR product LdRBLk obtained in this study by qPCR (**top line**). The positions of point nucleotide substitutions in the *LdRBLk* gene region obtained by qPCR are underlined and boxed. The box shows the amino acid residues of the hydrophilic center of the LdRBLk peptide.

**Figure 2 jof-07-00364-f002:**
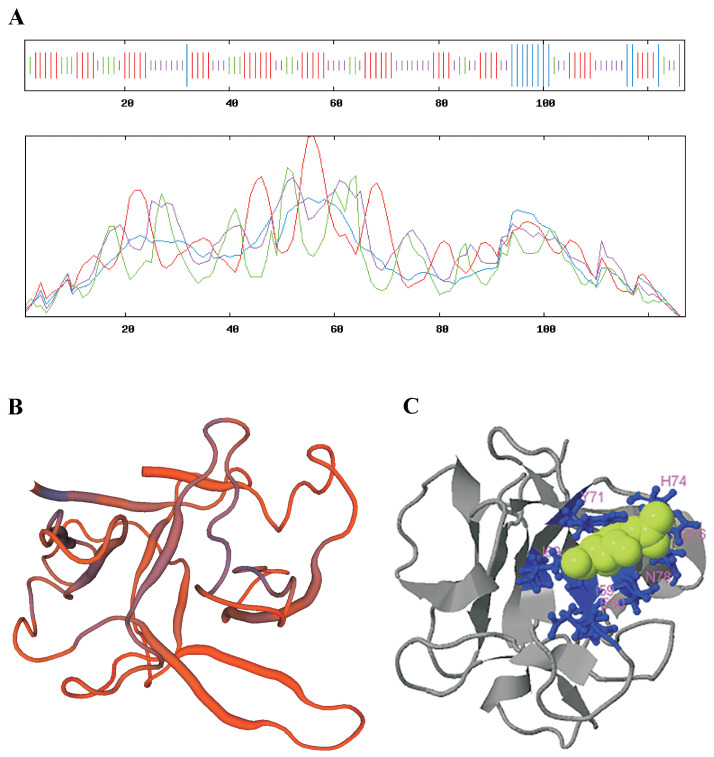
(**A**) Predicted secondary structure of the LdRBLk protein in the online SOPMA program. Strokes and lines are shown in blue, red, green and purple, representing an alpha helix, an extended strand, a beta turn and a random coil, respectively. Predicted tertiary structure of the LdRBLk peptide. (**B**) QMEAN color scheme constructed by SWISS-MODEL homology modeling. (**C**) Prediction of the ligand binding site for the LdRBLk peptide by the I-TASSER program.

**Figure 3 jof-07-00364-f003:**
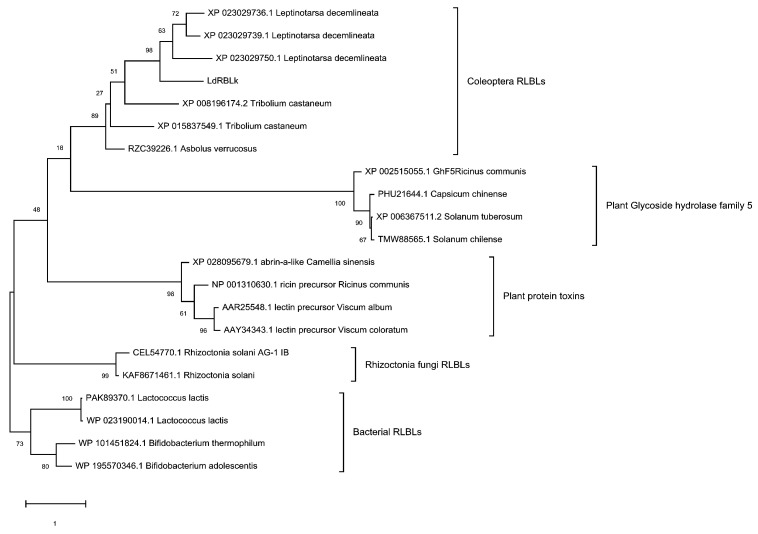
Evolutionary relationships of RBLs sequences from *L. decemlineata* and other Coleoptera, plant glycoside hydrolase family 5, plant protein toxins, *R. solani* fungi and bacteria inferred by using the Maximum Likelihood method and Whelan and Goldman + Freq. model [47]. The tree with the highest log likelihood (−9927.80) is shown. The percentage of trees in which the associated taxa clustered together is shown next to the branches. A discrete Gamma distribution was used to model evolutionary rate differences among sites (5 categories (+ G, parameter = 10.2324)). The tree is drawn to scale, with branch lengths being measured as the number of substitutions per site.

**Figure 4 jof-07-00364-f004:**
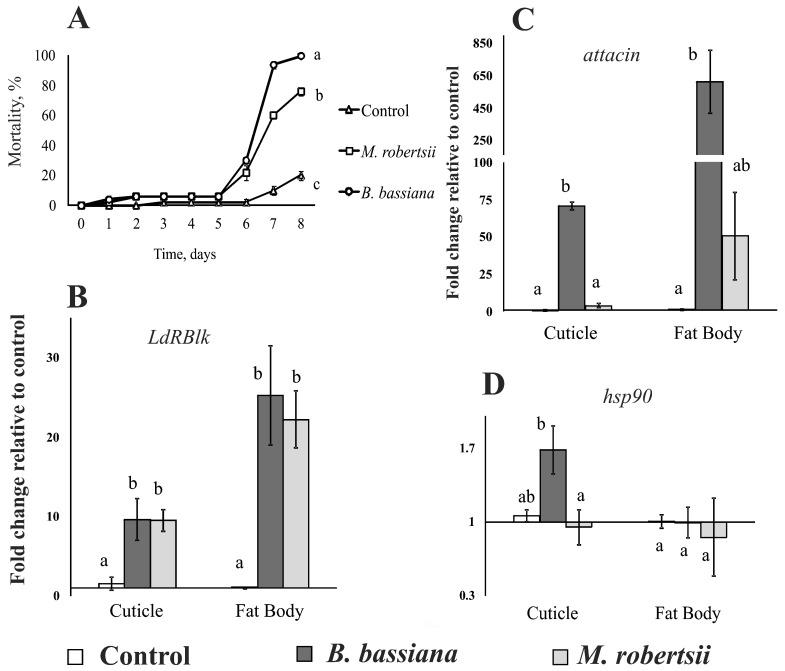
Mortality dynamics (**A**) and changes in the expression of *LdRBlk* (**B**), *attacin* (**C**) and *hsp90* (**D**) in the cuticle and fat body of Colorado potato beetle larvae after their dipping in suspensions of *B. bassiana* and *M. robertsii* conidia. The expression was measured 48 h after infection. Data normalized to reference genes: ribosomal protein L4 (*Rp4*) and ADP-ribosylation factor 1 (*Arf19*). The *Y*-axis shows the fold change relative to uninfected larvae, calculated separately for each tissue. Different letters show significant differences between treatments (log-rank test *χ*^2^ > 11.2, df = 1, *p* < 0.001 for mortality assay, and Dunn’s test, *p* < 0.05 for gene expression).

**Figure 5 jof-07-00364-f005:**
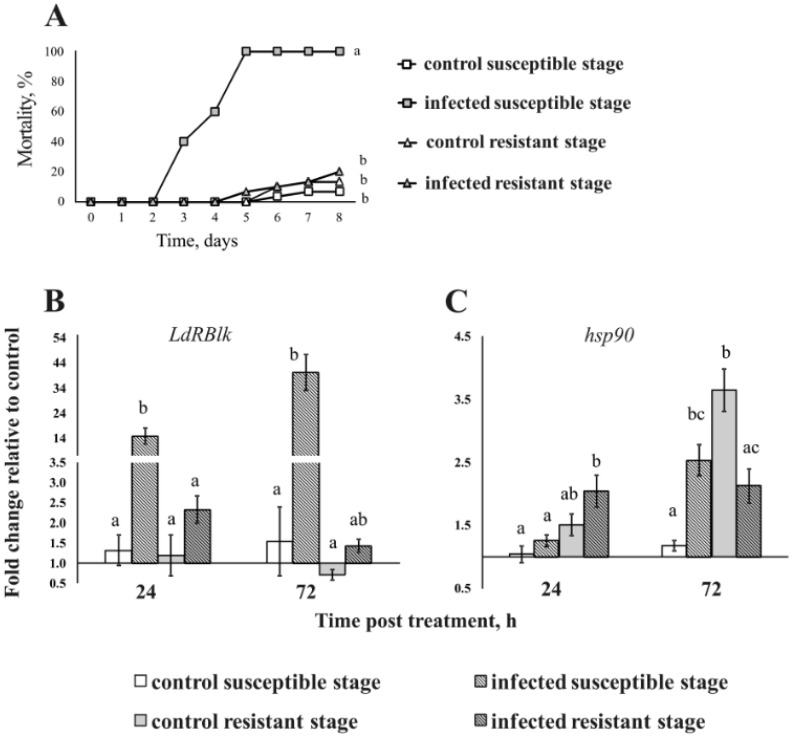
Mortality dynamics (**A**) and changes in the expression of *LdRBlk* (**B**) and *hsp90* (**C**) in the cuticle of susceptible stage (4 h post-molt in IV instar) and resistant stage (86 h post-molt in IV instar) Colorado potato beetle larvae after their dipping in a suspension of *M. robertsii* conidia. Expression was measured at 24 and 72 h after infection. Data normalized to reference genes: ribosomal proteins L4 and L18 (*Rp4* and *Rp18*) and ADP-ribosylation factors 4 and 1 (*Arf2* and *Arf19*). The *Y*-axis shows the fold change relative to uninfected larvae at 24 h. Different letters show significant differences between treatments (log-rank test *χ*^2^ < 55.8, df = 1, *p* < 0.001 for mortality assay, and Dunn’s test, *p* < 0.05 for gene expression).

**Figure 6 jof-07-00364-f006:**
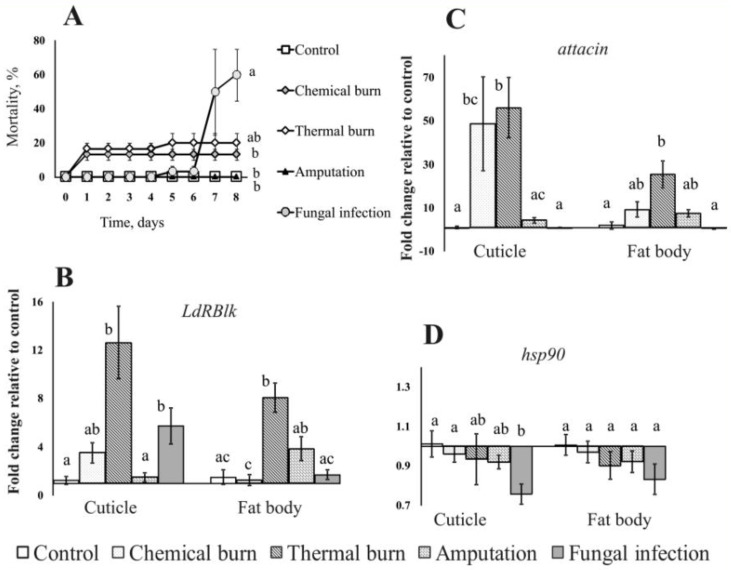
Mortality dynamics (**A**) and changes in the expression of *LdRBlk* (**B**), *attacin* (**C**) and *hsp90* (**D**) in the cuticle and fat body of Colorado potato beetle larvae after topical injuries of cuticle and topical treatment by fungal suspension (5 µL). The expression was measured at 24 h after treatments. Data normalized to reference genes: ribosomal proteins L4 and L18 (*Rp4* and *Rp18*) and ADP-ribosylation factor 1 (*Arf19*). The *Y*-axis shows the fold change relative to uninfected larvae, calculated separately for each tissue. Different letters show significant differences between treatments (log-rank test *χ*^2^ < 10.7, df = 1, *p* < 0.001 for mortality assay, and Dunn’s test, *p* < 0.05 for gene expression).

**Figure 7 jof-07-00364-f007:**
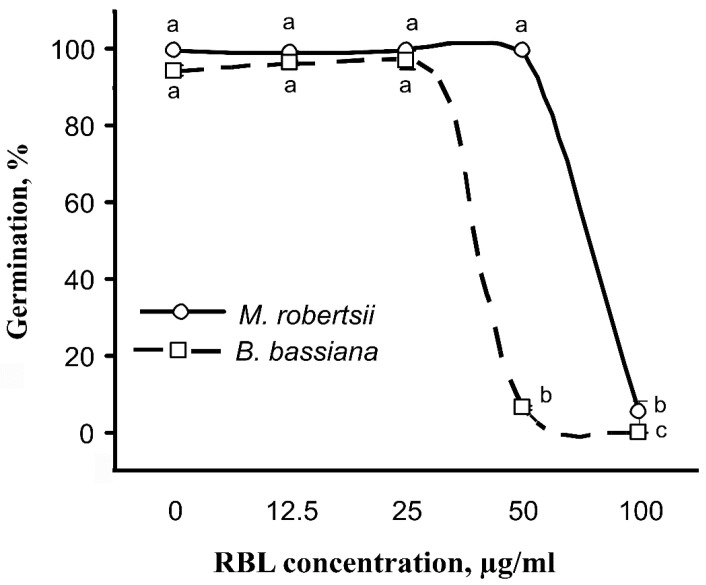
Effect of RBL *R. communis* on germination of *M. robertsii* and *B. bassiana* conidia in Sabouraud broth after 36 h of incubation. Different letters show significant differences between all treatments (Tukey’s test, *p* < 0.05).

## Data Availability

The data presented in this study are available on request from the corresponding authors.

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
