# Peer review of "Identification of the Ricin-B-Lectin LdRBLk in the Colorado Potato Beetle and an Analysis of Its Expression in Response to Fungal Infections"

_jof, 2021, doi:10.3390/jof7050364_

Round 1

Reviewer 1 Report

The paper was bioinformatic and functional analyzed the ricin-B-lectin LdRBlk, the data indicates the expression level of this gene depends on insect susceptibility to fungi and is correlated with the expression of other AMPs, such as attacin. Orthologous protein’s function analysis suggested the LdRBlk may able to suppress the development of the fungi B. bassiana and M. robertsii. The experiments are well designed and carry out. The data analysis is adequate and appropriate.

Comments:

1, Please give detail of the resistant insects. were they collected in the field or bred in the lab?

2, The bioinformatic analysis indicated the target ligand for LdRBLk is N-acetyl-D-galactosamine. The question is that is the orthologou protein from R. communis used in this paper have same target ligand? If the orthologou protein have same target ligand, the “Dose-dependent inhibition of conidial germination under the influence of RBL was observed for M. robertsii and B. bassiana (concentration effect – F4.20 = 6568.3, P < 0.001, Fig. 10)” experiment should test the N-acetyl-D-galactosamine effects on the conidial germination inhibition.

Reviewer 2 Report

The manuscript by Rotskaya and colleagues identified and analyzed the Ricin-B-lectin gene, LdRBLk, in the Colorado potato beetle in terms of their expression against fungal infections. This is a quite fundamental study that characterized the gene responding for fungi infections. It would be nice to add a bit more information/further study that you mentioned in Line 572-576.  

Here are some minor comments which will improve this manuscript.

Font and Size should be unified.

Too many figures. Some of them can be combined.

Line 13:  omit 78-89

Figure1 needs to be modified much further. Label on the left. I am quite concern that this should be the main figure. Or combine with Figure 6.

It is totally up to you but Figure 2,3 and 4 can be combined into a figure since they are indicating the protein structures.

Line 231: did you find the mutations from several beetle samples?  I guess “synonymous substitution” is a scientific terminology for this situation.

Line 258-262: Signal peptide is a part of the protein. Therefore, “LdRBLK contain/have a signal peptide“ would be the proper expression.

Line 321: If you use delta delta Ct and set the control as 1, I don’t see 3.5 to 3.8 fold changes in cuticles. It looks like the expression is close to 10-time fold changes.

Figure 7 & 9 should be formatted in a neat way. AND If you want to explain A -> C -> D -> B in the text, change the labels.  

Line 345-346: In general, I am curious about the susceptibility of adults.

Any reasons for checking expression only in the cuticle and the fat body?

Line 482: delete Nakmura.   

Round 2

Reviewer 1 Report

Although there is no particularly good answer to my question, the results of the paper are sufficient and worthy of publication.

Reviewer 2 Report

Great job!